# Quantitative Determination of Ethylene Using a Smartphone-Based Optical Fiber Sensor (SOFS) Coupled with Pyrene-Tagged Grubbs Catalyst

**DOI:** 10.3390/bios12050316

**Published:** 2022-05-10

**Authors:** Xin Yang, Justin Lee Kee Leong, Mingtai Sun, Linzhi Jing, Yuannian Zhang, Tian Wang, Suhua Wang, Dejian Huang

**Affiliations:** 1Department of Food Science and Technology, National University of Singapore, 2 Science Drive 2, Singapore 117542, Singapore; xinyang@u.nus.edu (X.Y.); justinleekl@gmail.com (J.L.K.L.); e0047517@u.nus.edu (L.J.); zhangyuannian@u.nus.edu (Y.Z.); 2School of Environmental Science and Engineering, Guangdong University of Petrochemical Technology, Maoming 525000, China; wangsuhua@ncepu.edu.cn; 3National University of Singapore (Suzhou) Research Institute, 377 Linquan Street, Suzhou 215123, China; 4Department of Chemistry, National University of Singapore, 3 Science Drive 3, Singapore 117543, Singapore; tian.wang91@u.nus.edu

**Keywords:** ethylene, fluorescence probe, smartphone, Grubbs catalyst, fruit ripening

## Abstract

For rapid and portable detection of ethylene in commercial fruit ripening storage rooms, we designed a smartphone-based optical fiber sensor (SOFS), which is composed of a 15 mW 365 nm laser for fluorescence signal excitation and a bifurcated fiber system for signal flow direction from probe to smartphone. Paired with a pyrene-tagged Grubbs catalyst (PYG) probe, our SOFS showed a wide linearity range up to 350 ppm with a detection limit of 0.6 ppm. The common gases in the warehouse had no significant interference with the results. The device is portable (18 cm × 8 cm × 6 cm) with an inbuilt power supply and replaceable optical fiber sensor tip. The images are processed with a dedicated smartphone application for RGB analysis and ethylene concentration. The device was applied in detection of ethylene generated from apples, avocados, and bananas. The linear correlation data showed agreement with data generated from a fluorometer. The SOFS provides a rapid, compact, cost-effective solution for determination of the fruit ethylene concentration dynamic during ripening for better fruit harvest timing and postharvest management to minimize wastage.

## 1. Introduction

Perishability of fresh fruits has been a long-standing supply chain issue, causing a sizeable proportion of the fruits to be discarded before distribution to the consumers [1,2]. Ethylene (CH_2_=CH_2_) is a major plant hormone that dictates fruit ripening and can trigger complex signaling pathways that control plant growth, development, adaptation to environmental stresses and pathogens [2,3,4,5,6,7,8,9,10,11,12,13]. Commercially controlling the ripening dynamics, by manipulation of ethylene concentration, of climacteric fruits has been used to extend shelf-life and ensure shelf-maturity. However excessive ethylene during storage or ripening can cause spoilage [2,14]. In the agricultural industry, the mechanisms of fruit ripening and spoilage have been well studied [2,4,15,16,17,18]. Trace concentration ethylene (1 ppm) has shown to initiate the ripening of climacteric fruits [16], while ethylene-sensitive fruits such as banana and kiwi have shown to be affected by sustained exposure to 10 ppb ethylene [19,20]. Therefore, ethylene concentration has been used as a maturity index to identify an optimal harvest period [5], define ideal storage conditions [21,22], and ripening speed [23,24].

Currently, commercial methods of ethylene detection based on gas chromatography (GC) [25], electrochemical methods [26], optical sensors [27], photoluminescence sensors [28], and laser-based sensors [25] are limited in practical application. GC and laser-based systems are hindered by price while a lack of selectivity limits effectiveness of electrochemical methods [2]. Chemical methods have been researched lately to make up for such shortfalls; however, few industrial-ready devices have been applied [2]. Such chemical methods include silver-impregnated poly(vinyl phenyl ketone) film based on the fluorescence quenching effects of metals [29]. Built upon metal complex quenching, a turn-on mechanism based on an ethylene competitive copper scorpionate complex conjugated with fluorescent polymer was developed for ethylene detection [30,31]. Kodera et al. introduced a reversible sensor for ethylene based on competitive d-π of ethylene with Ag(I) ion. In view of research into this domain, it was observed that such metal coordination-based methods under complex environments impose either a selectivity or sensitivity disadvantage. Recently, fluorophore-tagged Grubbs catalysts have displayed potential applicability with good selectivity [5,32]. Our group (2019) reported a novel pyrene-tagged first generation Grubbs catalyst (PYG) as a turn-on fluorescent probe for ethylene detection. The reported method showed good sensitivity (LOD of 0.9 ppm), selectivity against common interfering gases, and visible light emission suitable for smartphone detection. Michel et al. reported a Hoveyda–Grubbs second generation catalyst-based fluorescence probe with BODIPY as the fluorophore for the detection of ethylene [33]. Subsequently, taking advantage of its fast response, real-time monitoring, and high sensitivity, several fluorophore-tagged metal-complex-based fluorescence sensors were reported, showing great application potential for ethylene detection [34,35,36,37,38].

Herein, a smartphone-based optical fiber sensor (SOFS) was reported with high selectivity and wide linearity range (up to 350 ppm) and low detection limit (0.6 ppm). Moreover, an RGB-based image analysis software was developed for onsite data collection and real-time data analysis for the correlation of image *B*/*G* values and ethylene concentrations. Our device could be applied in commercial fruit ripening storage rooms for rapid and onsite determination of ethylene during the ripening process of fruits and help more fruit businesses cut their operation costs by reducing food waste.

## 2. Experimental Section

### 2.1. Materials and Reagents

All chemicals and reagents were purchased from the commercial sources (Sigma-Aldrich Chemical Co., Singapore) and used directly without further purification, unless specified. Ethylene and CO_2_ used were purchased from Chem-Gas Pte Ltd., Singapore. Dichloromethane and tetrahydrofuran (THF) were purified by distillation and were dried over molecular sieves before use. The pyrene-tagged Grubbs catalyst probe (PYG) and interfering gases (SO_2_, NO_2_, H_2_S, and NH_3_) were prepared according to our previously reported method [5].

Ultrapure water (18.2 MΩ cm) was obtained using a Millipore water purification system. All glassware was cleaned with ultrapure water and dried before use. Fluorescence measurements were recorded on an FS5 Spectrofluorometer (Edinburgh Instruments Ltd., Edinburgh, UK) analyzed with Fluoracle^®^.

### 2.2. Equipment Operation for Ethylene Determination

Before determination, the phone was installed into our SOFS device as shown in Figure 1. Then, after sample was mixed in PYG solution, the detector fiber connected to phone camara was submerged and the laser was activated by pressing the button beside the device (Figure 1C). The fluorescence signals of PYG were recorded by phone camera through the detector fiber and real time analysis was undertaken to give the concentration of ethylene. The kit consists of a 365 nm excitation source that was delivered into the target solution using outgoing fibers. The emission spectra from the solution were collected through the detection fiber and transmitted into the smartphone CMOS sensor. The signals were deconflicted using a bifurcated optical fiber structure. Alignment was achieved using precise 3D printing to map optical fiber output to a smartphone sensor. The 3D printed case of the SOFS has dimensions of 18 × 8 × 6 cm as shown in Figure 1a. The SOFS that was used for handheld testing displayed stray light noise interference and the method of ethylene testing had to be conducted in a semi-dark environment for noise reduction. Future designs of SOFS and probe vials would make use of suitable visible light absorbing materials to remove environmental lighting interference for a higher signal to noise ratio.

### 2.3. Design of Smartphone-Based Optical Fiber Sensor (SOFS)

The SOFS kit contains a 15mW excitation laser source (365 nm LED) (Zhuhai Photoelectric Co. Ltd., Zhuhai, China) powered by 12 V batteries, an optical fiber system (Zhuhai Photoelectric Co. Ltd., Zhuhai, China) linked to the smartphone, six vials of synthesized probe for ethylene determination, and spare replaceable filters. A smartphone (Samsung S8+, Seoul, Korea) with a CMOS S5K2L2 1.4 μm pixel RGB sensor was used for fluorescence signal collection and data analysis. The signals between sources were connected with a bifurcated connector that led to the fiber tip for sample data collection. A 3-D printed shell was fabricated to integrate the smartphone, laser, batteries, and optical fiber system.

### 2.4. Principle of Ethylene Determination

The reaction mechanism for ethylene determination is shown in Figure 2a. When PYG was exposed to ethylene, the fluorophore part from the probe was released and resulted in the fluorescence enhancement.

### 2.5. Determination of SOFS Coupled with PYG’s Sensitivity towards Ethylene

The process of ethylene determination is illustrated in Figure 2b according to our previous method. The PYG probe was dissolved in CH_2_Cl_2_ to obtain a stock solution with a concentration of 1 × 10^−3^ M. Ethylene gas of an initial concentration of 1000 ppm was diluted to 500, 400, 350, 300, 250, 200, 150, 100, 75, 50, 37.5, 25, 20, and 15 ppm by adding a known amount of ethylene gas to a round-bottom flask of a certain volume. Then, 10 mL of the corresponding gas were injected slowly into a test tube of 5 mL of CH_2_Cl_2_ solvent. A total of 1 mL of the sample solution was transferred to a small, sealed bottle before 30 μL of PYG probe stock solution were added. The solution was shaken for 3 min, performed in triplicate. Controls were prepared by the addition of 30 μL PYG probe stock solution to 1 mL CH_2_Cl_2_. A photograph of the solution was taken using an SOFS with the image settings ISO 800, exposure time 1/30 s, and 1440 p. The settings used were optimized by trial and error.

RGB values were extracted and processed using an android-based application written in Java programming language and developed in Android Studio Version 3.5 [39] powered by IntelliJ. An algorithm was established according to the relationship between the RGB values of the final solution and the ethylene concentration. The user interface of the app is shown in Figure 3. Sample preparation was repeated for spectrofluorometer analysis. The fluorescence emission spectra were recorded in the range of 380–600 nm at 365 nm excitation wavelength and a 0.1 s dwell time.

### 2.6. SOFS Conservation of PYG Selectivity

In general, interfering gases (SO_2_, NO_2_, H_2_S, CO_2_, and NH_3_ at 100 ppm) were bubbled into CH_2_Cl_2_ under the same conditions that were bubbled in ethylene gas. Similarly, the PYG probe stock solution (30 μL) was added and the image was captured for RGB analysis. Other coexisting species such as ac etonitrile, methanol, ethanol, ethyl acetate, water, toluene, THF, and 1-propanol were diluted in CH_2_Cl_2_ to 100 ppm before 30 μL of PYG stock solution was added.

### 2.7. Procedures for Smartphone Detection of Ethylene Released during Fruit Ripening

An airtight jar (Figure 2) was used to incubate apples (423 g) for 2, 4, 6, and 8 h. During each interval, sample gas of 2.0 mL was taken from the headspace of the jar using a gastight syringe with a long needle. The sample gas was bubbled slowly into 1.0 mL of CH_2_Cl_2_ solution in a capped vial to dissolve headspace gas. To this solution, the probe stock solution (30 μL) was added and the mixture was shaken for 3 min. RGB values were recorded using SOFS with image settings ISO 800, exposure time 1/30 s, and 1440 p. Ethylene released from the bananas (378 g) and avocados (435 g) was determined with the same procedures as that of apples. The concentration of ethylene in the jar (ppm) was measured using Equation (1) below:(1)$${\left[Ethylene\right]}_{ppm}=\left(\frac{B/G}{{\left(B/G\right)}_{0}}-1\right)/0.00213$$
where (*B*/*G*)_0_ and (*B*/*G*) represent the RGB values extracted from the SOFS sensor (λ_ex_ = 365 nm) of the blank and after the addition of ethylene, respectively.

The ethylene release rates of fruits tested was calculated using Equation (2) below:(2)$$R=\frac{\Delta_{ppm}V_{jar}}{W_{fruit}\Delta t}$$
where *R* is the ethylene release rate, Δ_*ppm*_ is the change in ethylene headspace concentration, *V_jar_* is the headspace of the jar, W_*fruit*_ is the weight of fruit tested, and Δ*t* is the amount time the fruit was incubated.

### 2.8. Statistical Analysis

Descriptive statistical analysis was performed using Origin 8.0 for calculating the means and the standard error of the mean. Results were expressed as the mean ± standard deviation (SD). 

## 3. Results and Discussion

### Design and Application of SOFS

The SOFS system hardware for ethylene concentration measurement is shown in Figure 1. Our SOFS kit includes an excitation laser source, an optical fiber system connected to the smartphone, a probe detection system, and a spare filter. The material of our detection fiber is silica. Our detection fiber has a wide wavelength range with good optical transparency. In the near-infrared spectral region, particularly around 1.5 μm wavelength, our fiber can have extremely low absorption and scattering losses of the order of 0.2 dB/km. A prototype of our SOFS is displayed in Figure 1.

The mechanism of ethylene determination and the step-by-step protocol for testing ethylene released from ripening fruits is shown in Figure 2. Using the ruthenium (Ru) based weakly fluorescent probe PYG, reported by us previously, fluorescence enhancement was observed via the release of fluorophore from the probe upon exposure to ethylene released from the fruit ripening process. The ethylene concentration could be determined through fluorescence enhancement of the probe which has a good dose–response relationship with the concentration of ethylene in a wide concentration range. A simple method was developed for determining the ethylene released from the fruit ripening process by enclosing fruits in an airtight jar. The ethylene releasing dynamics of the different fruits including avocado, banana, and apple were determined by monitoring the ethylene concentrations in the headspace of the jar over time.

An android application was developed for rapid analysis of RGB values shown in Figure 3. The android app made use of equation 1 for quantification of ethylene with RGB values.

*Sensitivity*. The feasibility of SOFS as a portable alternative to spectrofluorometers for ethylene detection was examined. Appendix A summarizes the solution RGB values at known concentrations of ethylene gas. Individual values R, G, or B displayed poor linear correlation with ethylene concentration, despite images captured showing a significant color shift from green to blue with increasing ethylene concentration based on visual observation as shown in Figure 4b. High variance in overall intensity was recorded despite efforts to standardize fiber tip positions. This may be caused by inherent angular misalignment that influences the intensity of light transferred due to the bending of the detection fiber tip [36]. Furthermore, smartphone individual RGB values cover overlapping ranges of wavelengths, that when analyzed separately could have resulted in large signal to noise ratio, thereby reducing the sensitivity of the test conducted.

Investigating the root cause of this shift, it was noted from the fluorescent emission spectra in Figure 4a that the intensity in the green region (peak at 503 nm) decreased gradually while intensity in the blue region (peak at 393 nm and shoulder at 415 nm) increased. The ratio between RGB values has been used to better describe the observation made in colorimetric assays [36]. The relationship between (*B*/*G*)/(*B*/*G*)_0_ values and concentration of ethylene was explored, as illustrated in Figure 4c. The (*B*/*G*)/(*B*/*G*)_0_ value of solution was shown to be linear from 0 to 350 ppm with an R^2^ value of 0.9947. These values depict a large linearity range with a good linear relationship with different concentrations of ethylene. The large range of linearity may be based on two reasons: first, our probe has a good sensitivity to even very low ethylene concentrations; and the second, silica optical fiber was applied to reduce the loss of optical signals during transmission.

The limit of detection (LOD) and limit of quantification (LOQ) were calculated using the equations LOD=3 σ/k and LOQ=10 σ/k, where σ is the standard deviation of the blank and k is the slope of the calibration line. The LOD and LOQ of ethylene were calculated to be 0.6 and 2 ppm, respectively. Although the LOD is approximately 3 times higher than when using a spectrofluorometer [5], the linearity range was adjusted to a larger range to cater for the higher concentrations of ethylene used in commercial ripening houses. At a 0.6 ppm LOD, it would be an adequately sensitive application for feedback control in such environments. Enrichment of the air in the fruit storage house may be undertaken to detect sub ppm level of ethylene. Furthermore, the compact size and low cost of the SOFS kit compared to that of the spectrofluorometer would enable constant onsite detection for fast correction.

*Selectivity and Interference*. Transition from a single wavelength monitoring to a spectra-analytical method was undertaken. Selectivity was investigated to determine if the PYG probe was maintained against other possible coexisting species. First, 100 ppm of ethylene was compared with a similar concentration of methanol, ethanol, 1-propanol, water, ethyl acetate, toluene, and tetrahydrofuran. It was noted that water, methanol, and ethanol showed slight differences in (*B*/*G*)/(*B*/*G*)_0_ values possibly due to slow decomposition of Grubbs catalyst [37,38]. However, this change was small when compared to the signal ratio resulting from ethylene as seen in Figure 5a. Potential interfering gaseous species in the ripening house, such as SO_2_, NO_2_, H_2_S, CO_2_, and NH_3_ were tested at 100 ppm as shown in Figure 5b. NH_3_ at 100 ppm induced a slight dimming of solution and decrease in (*B*/*G*)/(*B*/*G*)_0_ value while SO_2_ and CO_2_ showed slight increases in (*B*/*G*)/(*B*/*G*)_0_ values. These values coincide with our previously reported fluorescence, and no apparent interference was observed in fluorescence intensity which may be due to the high functional group tolerance of the Grubbs catalyst and high reactive priority of the vinyl group [5]. The SOFS was validated to maintain the PYG probe’s high selectivity and interference for application in complex warehouse conditions.

*Detection of Ethylene Released during Fruit Ripening*. During ripening, climacteric fruits display increased respiration and ethylene biosynthesis rates. Monitoring the ethylene release could provide insights into the maturity index of fruit. It worth considering that retail samples of fruits generally undergo an exogenous application of ethylene for better color and eating quality. However, poor control of ethylene could result in reduced shelf-life of fruits due to increased respiration rates. SOFS coupled with PYG provides the avenue to bring down the cost of rapid and onsite testing of ethylene. This could provide convenient testing to ensure quality control on both the wholesale and the retail levels. To test for this practical applicability, three climacteric fruits’ ethylene production during ripening was characterized. The ethylene releasing rates were calculated from ethylene concentrations monitored over time. The ethylene response curve of fruits over time is shown in Figure 6a and ethylene release rates shown in Figure 6b. The ethylene release rates of fruit calculated with Equation (2) above ranged from 17 to 46 μL kg^−1^ h^−1^. The banana displayed the highest rate of ethylene release out of the three tested, followed by avocados then apples. A higher ethylene release rate, based on the species, could be indicative of riper fruit. Fruits measured were also found to show a linear increase in ethylene concentration with increasing storage time. This displays the steady state biosynthesis and release of ethylene during storage.

## 4. Conclusions

In summary, we developed an SOFS that could be coupled well with a PYG probe to provide a comparable method for portable, rapid, and sensitive detection of ethylene. The detection limit for our system of 0.6 ppm is sufficient for commercial use with an R^2^ value of 0.9947, suggesting good reliability. A large linearity range could prove useful for high ethylene doses. It maintained the PYG probe’s selectivity despite using full spectrum data over single wavelength monitoring. In all, our SOFS sensor provides a user-friendly solution for ethylene detection with high efficiency and is of great importance for the real-time monitoring of fruits in storage and transport stages.

## Figures and Tables

**Figure 1 biosensors-12-00316-f001:**
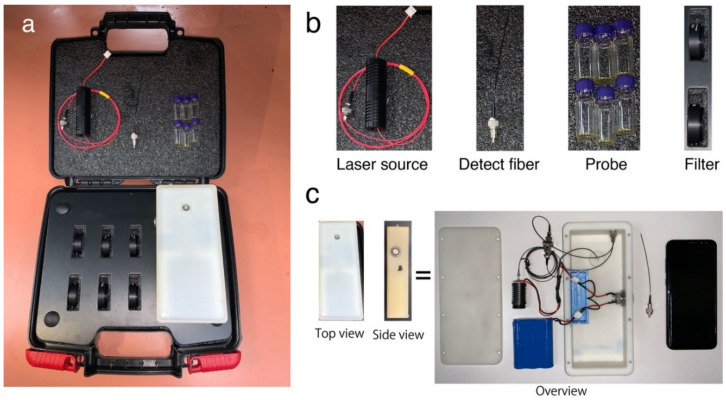
Illustration of an SOFS kit for the determination of ethylene concentration in air. (**a**) Overview of the SOFS kit. (**b**) Components of the SOFS kit. (**c**) Illustration of internal components consisting of battery, excitation source, bifurcated optical fiber connector, replaceable fiber optic and smartphone.

**Figure 2 biosensors-12-00316-f002:**
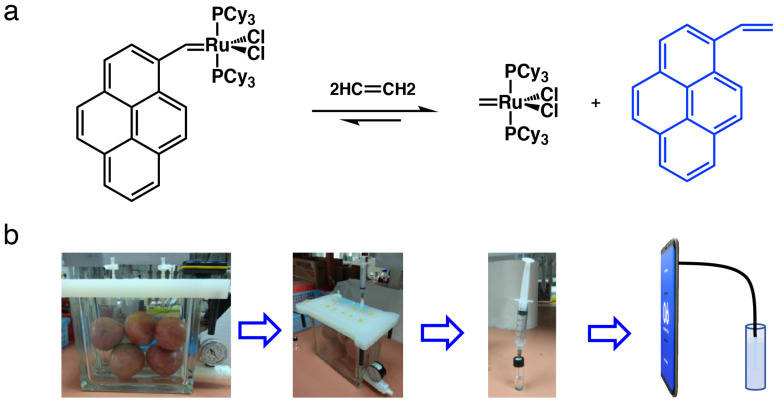
(**a**) Ethylene detection mechanism of the PYG probe; (**b**) schematic steps for ethylene detection using SOFS coupled with PYG.

**Figure 3 biosensors-12-00316-f003:**
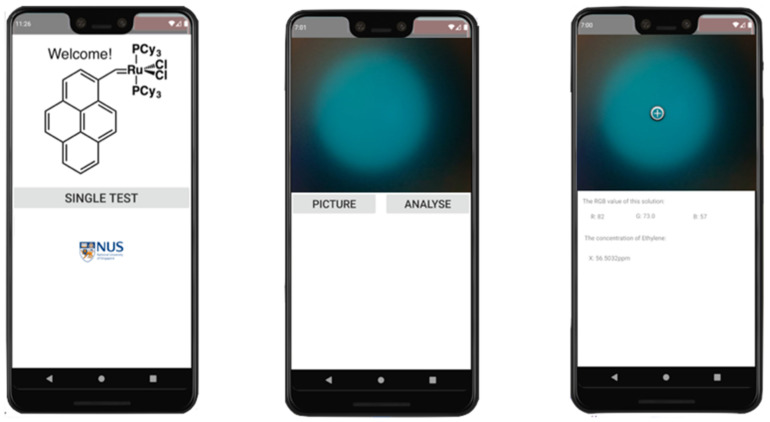
User interface of the android-based application with two functions: photo retrieval and RGB analysis.

**Figure 4 biosensors-12-00316-f004:**
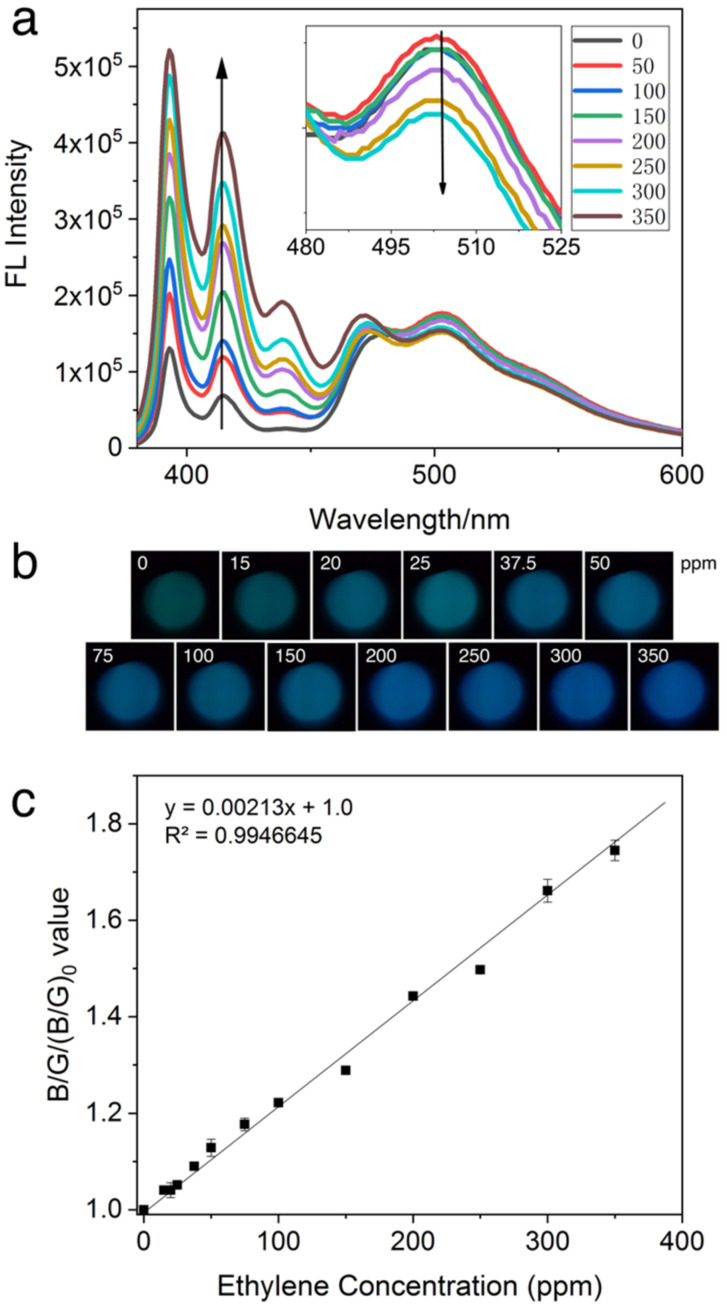
(**a**) Emission spectra of the PYG probe at 30 μM in CH_2_Cl_2_ exposed to 0–350 ppm ethylene, (**b**) the visual color changes of PYG corresponding to different concentrations of ethylene, and (**c**) plots of (*B*/*G*)/(*B*/*G*)_0_ value versus different concentrations of ethylene. Where (*B*/*G*)_0_ and (*B*/*G*) represent the RGB values extracted from CMOS sensor (λ_ex_ = 365 nm) of blank and after the addition of ethylene, respectively. The number of data points for the single determination is 3.

**Figure 5 biosensors-12-00316-f005:**
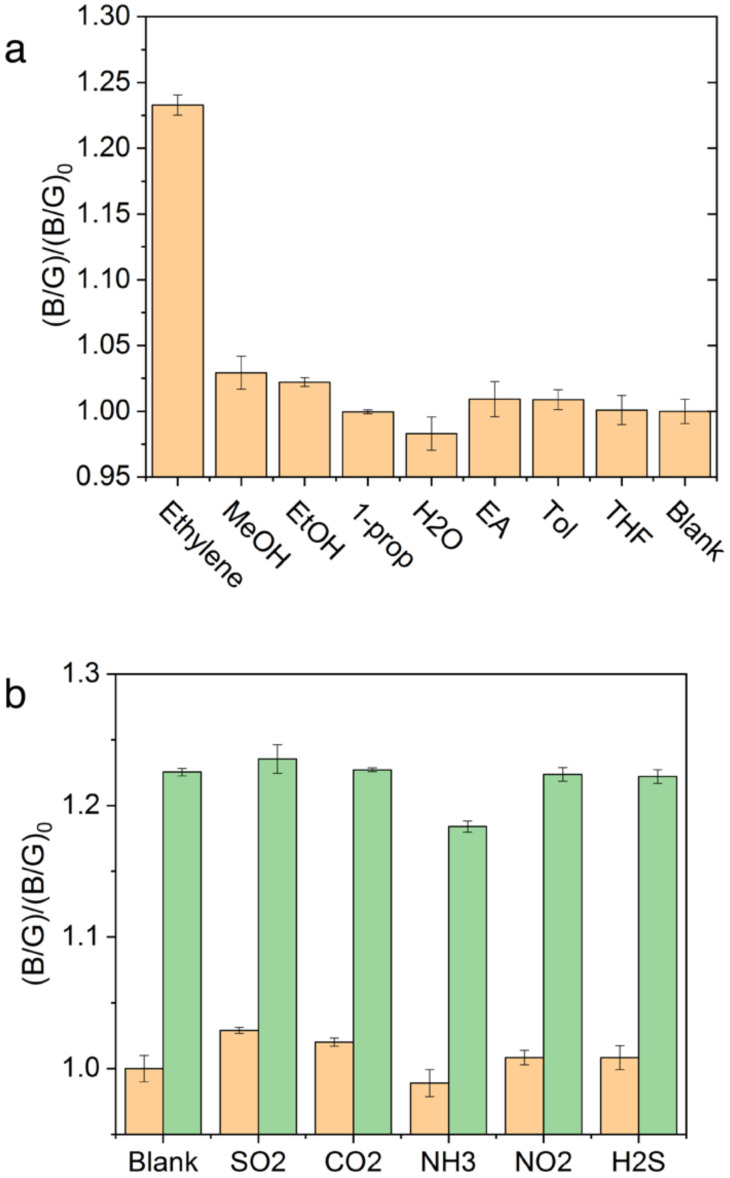
Selectivity and interference of PYG for ethylene in the presence of other (**a**) liquid and (**b**) gaseous species at 100 ppm with (orange bar) and without the addition of ethylene (green bar).

**Figure 6 biosensors-12-00316-f006:**
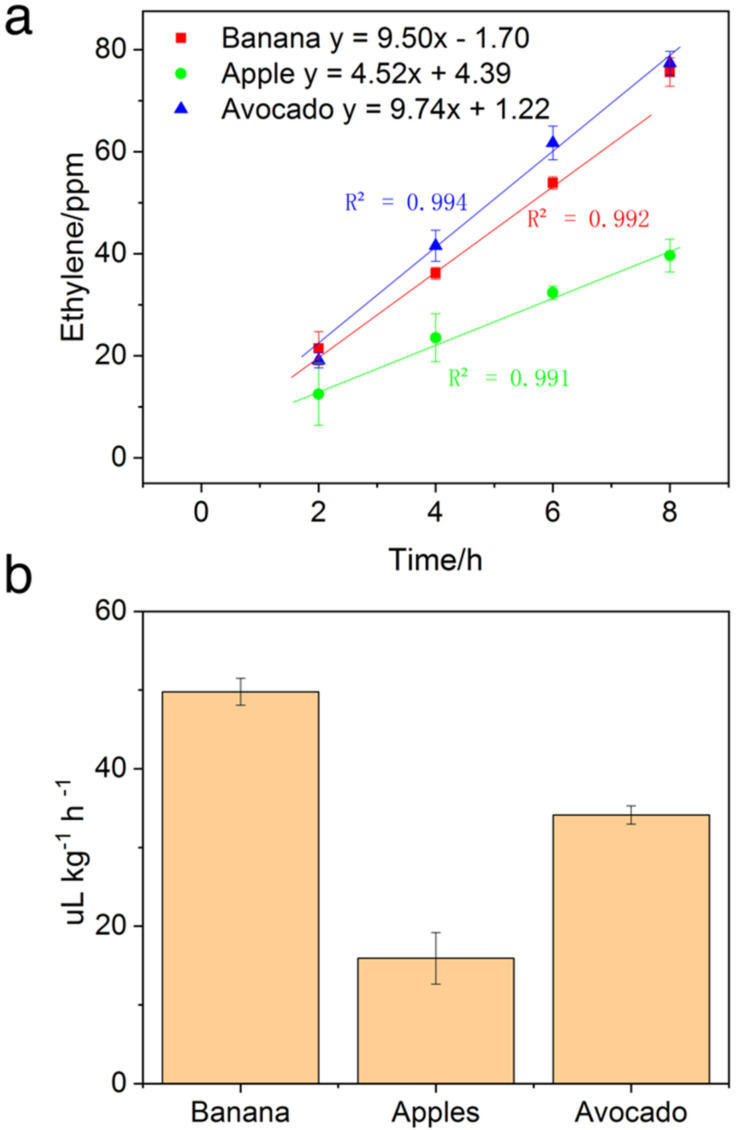
(**a**) SOFS detected ethylene release by fruits during the ripening process at different incubation periods. (**b**) Ethylene releasing rates of different fruits over 8 h.

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
