# Peer review of "Quantitative Determination of Ethylene Using a Smartphone-Based Optical Fiber Sensor (SOFS) Coupled with Pyrene-Tagged Grubbs Catalyst"

_biosensors, 2022, doi:10.3390/bios12050316_

Round 1

Reviewer 1 Report

The manuscript proposed an Ethylene detection based on a smartphone and a fiber-optic fluorescence sensor. The fluorescent signals generated by chemical changes are collected and transduced by commercial fiber-optic sensors, and the CCD of a smartphone can be thought of as a signal demodulation device. The method of dividing the intensity of two wavelengths (corresponding to the G and B photosensitive elements of the smartphone) for the broadband spectra is called differential demodulation, which is a common method used in the field of sensor demodulation, and the smartphone camera is only one of the specific implementation ways. We noticed that the manuscript used the same method to demodulate the sensing signal of the colorimetric reader in Ref. 37 (Sensors and Actuators B: Chemical 283 (2019): 524-531), except for the improvement of fluorescent molecules, the novelty of the manuscript is insufficient.

In addition, although this demodulation method has been confirmed in the application of colorimetric reader, it is not clear in fluorescence testing, for example, if the B/G response of two mixed gases is the same as the ethylene gases, can they be identified? How accurate the cell phone camera's measurement is, will it be affected by the conditions in different gas collection areas, as the amount of light passing through the fiber may vary. The manuscript needs sufficient experimental verification.

Moreover, the manuscript is unclear when discussing the contrasting experiments for interfering gases. Figure 5b lacks annotations for the histograms of the two colors and is not explained clearly in the manuscript. Fluorescence will also have the problem of false positives and noises, like the blank group in 5b, there will still be output values, and the average value of NH3 is smaller than that of the blank group, which indicates that the fluorescence signal is very weak. Whether the smartphone cameras can respond to weak fluorescence signals is debatable.

In general, the manuscript is not suitable for publication on biosensors

Author Response

The manuscript proposed an Ethylene detection based on a smartphone and a fiber-optic fluorescence sensor. The fluorescent signals generated by chemical changes are collected and transduced by commercial fiber-optic sensors, and the CCD of a smartphone can be thought of as a signal demodulation device. The method of dividing the intensity of two wavelengths (corresponding to the G and B photosensitive elements of the smartphone) for the broadband spectra is called differential demodulation, which is a common method used in the field of sensor demodulation, and the smartphone camera is only one of the specific implementation ways. We noticed that the manuscript used the same method to demodulate the sensing signal of the colorimetric reader in Ref. 37 (Sensors and Actuators B: Chemical 283 (2019): 524-531), except for the improvement of fluorescent molecules, the novelty of the manuscript is insufficient.

Reply: Thank you for the comment! We used the same platform developed in our lab for data collection and analysis, and this will save the experimental costs and reduce the chance of making mistakes. In the previous work, we proposed the idea of the whole detection system based on smartphones, including the principle of sensing system and color extraction, etc. Herein, we use 3D printing technology to further optimize and combine the whole detection system and build simple detection equipment. At the same time, optical fiber is used to further improve the accuracy and reliability of detection.  For this manuscript, the thing we want to highlight is not our platform but the solution that was put forward for the determination of ethylene with low cost and high efficiency.

In addition, although this demodulation method has been confirmed in the application of colorimetric reader, it is not clear in fluorescence testing, for example, if the B/G response of two mixed gases is the same as the ethylene gases, can they be identified? How accurate the cell phone camera's measurement is, will it be affected by the conditions in different gas collection areas, as the amount of light passing through the fiber may vary. The manuscript needs sufficient experimental verification.

Reply: Thank you for the comment! Our probe have good functional group tolerance and high reactive priority to vinyl group, so only gases with vinyl group could have fluorescence response. Moreover, our method was designed to determine the ethylene concentration in the ripening house and the potential interfering gaseous species have been proved not affect the test result. Our method shows good linear relationship between (B/G)/(B/G)0 value and different concentrations of ethylene just as showed in Figure 4C.

Moreover, the manuscript is unclear when discussing the contrasting experiments for interfering gases. Figure 5b lacks annotations for the histograms of the two colors and is not explained clearly in the manuscript.

Reply: Thank you for the comment! We have provided the explanation of interfering gases and the annotations for the histograms of the two colors in the revised manuscript.

Fluorescence will also have the problem of false positives and noises, like the blank group in 5b, there will still be output values, and the average value of NH3 is smaller than that of the blank group, which indicates that the fluorescence signal is very weak. Whether the smartphone cameras can respond to weak fluorescence signals is debatable.

Reply: Thank you for the comment! The deducted fluorescence in the presence of NH3 is due to the decomposition of the Grubbs catalyst induced by NH31, so it is not a false positive. And the smartphone fluorescence sensors have been widely applied in the chemicals analysis2-4, the LOD could be optimized to satisfy our requirements.

  1. Vougioukalakis, G. C.; Grubbs, R. H., Ruthenium-based heterocyclic carbene-coordinated olefin metathesis catalysts. Chem. Rev. 2010, 110 (3), 1746-1787.
  2. Shen, Y.; Wei, Y.;  Zhu, C.;  Cao, J.; Han, D.-M., Ratiometric fluorescent signals-driven smartphone-based portable sensors for onsite visual detection of food contaminants. Coord. Chem. Rev. 2022, 458, 214442.
  3. McCracken, K. E.; Yoon, J.-Y., Recent approaches for optical smartphone sensing in resource-limited settings: a brief review. Analytical Methods 2016, 8 (36), 6591-6601.
  4. Zhao, W.; Tian, S.;  Huang, L.;  Liu, K.;  Dong, L.; Guo, J., A smartphone-based biomedical sensory system. Analyst 2020, 145 (8), 2873-2891.

Reviewer 2 Report

Article “Quantitative Determination of Ethylene Using Smartphone 2 Based Optical Fiber Sensor (SOFS) coupled with Pyrene-tagged Grubbs Catalyst” is devoted e designed a smartphone based optical fiber sensor (SOFS), which is composed  of a 15 mW 365 nm laser for fluorescence signal excitation and a bifurcated fiber system for signal  flow direction from probe to smartphone. The results obtained in the work can find important practical application. The article contains interesting results and corresponds to the theme of the journal Biosensors.

Nonetheless, to improve the article, I recommend that the authors pay attention to the following comments:

  1. I recommend that the authors clearly state the purpose of the work in the final part of the introduction.
  2. Equation (1) should be deciphered.
  3. How can the authors explain the large range of linearity of the analytical signal obtained using the developed sensor?
  4. What was the material and optical characteristics of the detect fiber?

I think that after a minor revision, the article can be published in the Biosensors journal.

Author Response

Article “Quantitative Determination of Ethylene Using Smartphone 2 Based Optical Fiber Sensor (SOFS) coupled with Pyrene-tagged Grubbs Catalyst” is devoted e designed a smartphone based optical fiber sensor (SOFS), which is composed  of a 15 mW 365 nm laser for fluorescence signal excitation and a bifurcated fiber system for signal  flow direction from probe to smartphone. The results obtained in the work can find important practical application. The article contains interesting results and corresponds to the theme of the journal Biosensors.

Nonetheless, to improve the article, I recommend that the authors pay attention to the following comments:

  1. I recommend that the authors clearly state the purpose of the work in the final part of the introduction.

Reply: Thank you for the comment! The purpose of the work has been provided in the revised final part of the introduction to give a better understanding of our work.

  1. Equation (1) should be deciphered.

Reply: Thank you for pointing out! The equation (1) has been deciphered.

  1. How can the authors explain the large range of linearity of the analytical signal obtained using the developed sensor?

Reply:

Our probe has a good sensitivity to even very low ethylene concentration and we apply silica optical fiber to reduce the loss of optical signals during transmission. All of these ensure that our sensor has a large linear range.

  1. What was the material and optical characteristics of the detect fiber?

Reply: The material of our detect fiber is silica. Our detect fiber has a wide wavelength range with good optical transparency. In the near-infrared spectral region, particularly around 1.5 μm wavelength, our fiber can have extremely low absorption and scattering losses of the order of 0.2 dB/km.

I think that after a minor revision, the article can be published in the Biosensors journal.

Round 2

Reviewer 1 Report

The revised manuscript has explained the questions we posed, and it is recommend to be published on Biosensors.